# Effectiveness of a multi-country implementation-focused network on quality of care: Delivery of interventions and processes for improved maternal, newborn and child health outcomes

Nehla Djellouli[1]*, Yusra Ribhi Shawar[2,3], Kasonde Mwaba[1], Kohenour Akter[4], Gloria Seruwagi[5], Asebe Amenu Tufa[6], Geremew Gonfa[6], Kondwani Mwandira[7], QCN Evaluation Group[¶], Agnes Kyamulabi[5], Jeremy Shiffman[2,3], Mike English[8], Tim Colbourn[1]*

1 Institute for Global Health, University College London, London, United Kingdom, 2 Bloomberg School of Public Health, John Hopkins University, Baltimore, Maryland, United States of America, 3 Paul H. Nitze School of Advanced International Studies, John Hopkins University, Washington, DC, United States of America, 4 Perinatal Care Project, Diabetic Association of Bangladesh, Dhaka, Bangladesh, 5 School of Public Health, Makerere University, Kampala, Uganda, 6 Ethiopian Public Health Institute, Addis Ababa, Ethiopia, 7 Parent and Child Health Initiative PACHI, Lilongwe, Malawi, 8 Centre for Tropical Medicine and Global Health, University of Oxford, Oxford, United Kingdom

¶ Membership of QCN Evaluation Group is provided in the Acknowledgments
* n.djellouli@ucl.ac.uk (ND); t.colbourn@ucl.ac.uk (TC)

**Data Availability Statement:** All data is derived from qualitative interviews, most with stakeholders

## Abstract

The Network for Improving Quality of Care for Maternal, Newborn and Child Health (QCN) aims to work through learning, action, leadership and accountability. We aimed to evaluate the effectiveness of QCN in these four areas at the global level and in four QCN countries: Bangladesh, Ethiopia, Malawi and Uganda. This mixed method evaluation comprised 2–4 iterative rounds of data collection between 2019–2022, involving stakeholder interviews, hospital observations, QCN members survey, and document review. Qualitative data was analysed using a coding framework developed from underlying theories on network effectiveness, behaviour change, and QCN proposed theory of change. Survey data capturing respondents' perception of QCN was analysed with descriptive statistics. The QCN global level, led by the WHO secretariat, was effective in bringing together network countries' governments and global actors via providing online and in-person platforms for communication and learning. In-country, various interventions were delivered in 'learning districts', however often separately by different partners in different locations, and pandemic-disrupted. Governance structures for quality of care were set-up, some preceding QCN, and were found to be stronger and better (though often externally) resourced at national than local levels. Awareness of operational plans and network activities differed between countries, was lower at local than national levels, but increased from 2019 to 2022. Engagement with, and value of, QCN was perceived to be higher in Uganda and Bangladesh than in Malawi or Ethiopia. Capacity building efforts were implemented in all countries–yet often dependent on implementing partners and donors. QCN stakeholders agreed 15 core monitoring indicators

where only one individual holds a position, either within federal or state government, facilities, or NGOs. Every care has been taken to ensure anonymity of the data in the submitted manuscript but the authors from all 4 countries feel strongly that making data freely available would jeopardise the conditions of informed consent. We have uploaded a detailed methods document within the Supporting Information files. Data requests may be sent to the UCL data protection office: data-protection@ucl.ac.uk and UCL ethics committee ethics@ucl.ac.uk.

**Funding:** This work was funded by the Medical Research Council (MRC) Health Systems Research Initiative 5th call via grant MR/S013466/1 to TC at UCL Institute for Global Health, United Kingdom, YS and JS at Johns Hopkins University, United States of America, KA and AK at Diabetic Association of Bangladesh Perinatal Care Project, Bangladesh, CM at Parent and Child Health Initiative, Malawi, GS at Makerere University School of Public Health, Uganda, and ME at University of Oxford, United Kingdom; and by the Bill & Melinda Gates foundation via grant INV-007644 to TM at LSHTM, United Kingdom. The funders had no role in study design, data collection and analysis, decision to publish, or preparation of the manuscript.

**Competing interests:** The authors have declared that no competing interests exist

though data collection was challenging, especially for indicators requiring new or parallel systems. Accountability initiatives remained nascent in 2022. Global and national leadership elements of QCN have been most effective to date, with action, learning and accountability more challenging, partner or donor dependent, remaining to be scaled-up, and pandemic-disrupted.

## Introduction

Improving the quality of care is an increasing focus of global health [1–3], and is high on the global maternal, newborn and child health (MNCH) agenda [4]. In February 2017, the Network for Improving Quality of Care for Maternal, Newborn and Child Health (QCN) was launched in Lilongwe, Malawi [5]. QCN brings together government ministries of health in 11 participating countries, implementing partner organisations and donors, to work with healthcare professionals to improve the quality of care for mothers, newborns and children, via work at global, national and local levels [6,7]. QCN's vision is "every mother and newborn receives quality care throughout pregnancy, childbirth and postnatal period" through values of "quality, equity and dignity" and its stated goal is "halving maternal and newborn deaths in health facilities in five years" [8]. To achieve this goal, QCN agreed on four strategic objectives: Leadership, Action, Learning and Accountability (LALA), used as a framework to operationalise implementation of the network activities [8]. As detailed in the QCN strategic objectives document [8], leadership entails setting up policies, strategies and structures within countries to support the systems change needed to improve quality of care. The action strategic objective is focused on accelerating action through better coordination and harmonisation of efforts to improve quality of care, using evidence-based standards and interventions, as well as putting the learning of the network into practice [8]. Learning occurs at the national and global levels around what is needed to improve quality of care and how to achieve this, sharing best practice knowledge within and between countries [8]. Transparent data collection and documentation are required for learning, and for accountability, which also involves mechanisms to ensure community engagement and accountability for good user experience of care [8]. Each strategic objective outlines 3–4 outputs that network countries are expected to achieve [8]. Countries used this framework for the planning and programming of QCN activities in-country. WHO used the outputs laid out in the LALA framework to monitor countries in terms of implementation milestones [9].

This is the third paper in a collection evaluating the QCN (S1 Text). This paper looks at delivery of interventions in QCN at the global, national and local levels, which is of key importance given QCN is an implementation-focused network. The aim of this paper is to investigate QCN effectiveness, focusing on the work of the network at global level as well as in Bangladesh, Ethiopia, Malawi and Uganda–four of the 11 QCN countries we use as case studies—reflecting a range of QCN emergence, institutionalisation and embeddedness [7,10]. In this analysis, we therefore concentrate on the effectiveness of the network in changing processes that should improve maternal, newborn and child health outcomes.

This evaluation explores QCN's effectiveness by examining its outputs, policy consequences and impact [11]. Here we conceptualise outputs of the network as QCN activities from global to local levels, including development of quality of care standards for MNCH, international and national meetings, and quality improvement (QI) interventions. Policy consequences in our analysis focus on global and national policy processes such as the development of MNCH

quality of care roadmaps in each country, alignment of goals and funding between MNCH partners, and the scale-up of QI interventions in-country. As Shiffman et al. (2016) denotes, the impact of a network, particularly on improving population health is complex and difficult to determine due to a myriad of factors contributing to population health, within and beyond the efforts of the network [11]. Although the aim of the network was to reduce case fatality rates by 50% by the end of 2022, the measurement of that outcome is incomplete, and is beyond the scope of our research. Thus, we focus on intermediate processes and outputs of the network at all levels of governance. We take into consideration multiple influences within and beyond the network's activity that shape its effectiveness. We end by discussing the way in which the network's features and the policy environment shape QCN's effectiveness.

Understanding the effectiveness of QCN is important given its scale and ambition and the investments it has involved over five years (2017–2022). It is also crucial given QCN's possibility to influence the way international health organisations and donors operate in the future. This evaluation provides useful knowledge for future multi-country global health networks.

## Methods

### Ethics statement

All interviews, observations and surveys were conducted after obtaining written informed consent. Ethical approval was received from University College London Research Ethics Committee (ref: 3433/003); BADAS Ethical Review Committee (ref: BADAS-ERC/EC/19/00274), Ethiopian Public Health Institute Institutional Review Board (ref: EPHI-IRB-240-2020), National Health Sciences Research Committee in Malawi (ref: 19/03/2264) and Makerere University Institutional Review Board (ref: Protocol 869).

To evaluate the work of the QCN, we conducted multiple embedded case studies [12] in four countries–Bangladesh, Ethiopia, Malawi and Uganda–chosen for the spectrum of contexts they represented. This research design allowed us to investigate in more depth factors that might influence the effectiveness of the network by conducting a cross-country analysis of our case studies [12,13]. Each case study comprised embedded units of analysis: the national, sub-national and local levels, as well as the country's interactions with the global level of the QCN. Within each unit of analysis, we further delved into and triangulated a multitude of perspectives (such as governments, implementing partners, health facility workers and other partners of the network) and data sources (detailed below) since the inception of the network in 2016 until March 2022. We describe the case study settings in more detail in S2 Text.

To evaluate how the network operated, we conducted in multiple rounds: interviews with key QCN actors across different levels of the health systems; observations of QCN meetings and learning health facilities; a quantitative survey to understand QCN actors' awareness and perceptions of the network in each country; and analysis of documents related to QCN operationalisation and implementation. Semi-structured interviews were conducted with QCN actors at the global, national and local levels and included participants from diverse backgrounds and affiliations: members of the Ministry of Health (MoH), implementing partners (e.g. UNICEF, USAID), health facility workers (managerial and clinical staff), technical partners, academic partners, WHO and other global partners. We sought to pay particular attention to the perspectives and goals of those carrying out the work of the network [14,15]. In total, 248 interviews were conducted–including follow-up interviews of key stakeholders–over several rounds of data collection between March 2019 and March 2022, across all levels of governance in our four case study countries, and globally (see S2 Text for breakdown of interviews per case study and round of data collection). To accommodate local restrictions related to the global pandemic, some of the interviews at global and national levels were conducted online or

by telephone [16]. Initial topic guides were developed for global, national and local stakeholders that were then adapted to each country's context and translated in local languages as appropriate. Topic guides were further refined for each round of data collection to validate emerging findings; follow-up on QCN progress and emerging lines of inquiry; and to be more specific for the category of stakeholders. Interviews were conducted by several members of the QCN Evaluation Group trained in qualitative methods and familiar with the local contexts and languages. Interviews were then transcribed verbatim and translated into English, if not conducted in English. Interviews were complemented by non-participant observations [17] of international QCN meetings as well as national and sub-national QCN meetings in case-study countries. QCN activities were further observed in each country via visits to two best and two least performing QCN health facilities that were selected based on maternal and newborn health outcomes and other quality of care data used in national schemes relevant for each country. Non-participant observations served several purposes, such as: anchoring the case studies in the real-world setting of the cases [12]; exploring why QCN activities are done (or not) and how those changed over time; probing for any effects and the veracity of QCN monitoring data; and triangulating the data [17]. Observations were completed over several rounds of data collection between March 2019 and March 2022 (see S2 Text for breakdown of observations per case study) by members of the QCN Evaluation Group trained in qualitative methods and familiar with the local setting (culture, language, context). We used templates (see S2 Text) to capture key processes relevant to the focus of the network in each country during observations, as well as unstructured field notes. The focus of observation was sharpened over time enabled by iterative rounds of data analysis and reflection.

Additionally, we adapted a psychometrically validated tool (5 domains, 40 indicators) developed for evaluating clinical networks [18] to evaluate the network at national and local levels in each case study. Over several rounds, we surveyed a variety of network members (e.g. clinicians, managers, advisors) that also included QCN actors beyond our observation sites, totalling 1525 responses across the countries and rounds of data collection (see S2 Text for a breakdown per case study). Respondents had an option to fill in the survey online, via the Opinio platform, or on paper. Finally, we triangulated the data collected with a document review that included all relevant published and unpublished documents and communications relating to QCN at the global, national and district levels in the case study countries. These included strategy and management documents, operational plans, directives, formal minutes, and reports (see S2 Text). We were able to access unpublished documents via WHO and Ministry of Health QCN contacts.

All qualitative data was analysed using a common coding framework developed from several underlying theories that framed the overall QCN evaluation (see S2 Text). In this paper, our analysis was guided by the QCN Theory of Change and monitoring framework [8]–the Leadership, Action, Learning and Accountability (LALA) strategic objectives of QCN–and the environment, structure, process and outcomes of the QCN [19]. All data was coded in NVivo 12, drawing on initial theory in both an inductive and deductive way [17]. Our codebook contained 'theory' codes related to underlying theories; each theory was outlined using codes and sub-codes that broke down the different components of the theory. The codebook was further supplemented by 'case study' codes to distinguish data specifically relevant to each case study. We describe in more detail how the codebook was developed, piloted and tested by researchers from the QCN Evaluation Group in S2 Text. We worked to ensure consistency in our qualitative data collection and analysis via holding regular interactive group training sessions with all researchers in the team, led by senior members of the team. Over two years, many of our research team, including nine co-authors for this paper, were involved in coding data in the different case studies. Six of the co-authors were also actively involved in local data collection

and familiar with the local context. Regular team meetings and rechecks took place during each round of coding in order to ensure inter-coder consistency and that coders remained close to the theories underpinning the codebook. Team meetings were also an opportunity for coders to put forward new codes they deemed relevant to refine the 'theory' codes or to address gaps in 'case study' codes. The quantitative data was analysed independently in the first place, by conducting descriptive statistics of respondents and the percentages of respondents giving responses to each question. In S2 Text, we provide a more detailed explanation of the quantitative data analysis process. After analysis, the results of the survey were triangulated with the qualitative data.

QCN Evaluation group members who were from, and based in, the countries being researched, conducted the interviews in each country. They were experienced qualitative researchers familiar with the culture, language, context and health system in the country, though not working in the health system or the particular health facilities involved. As such they were non-participant observers and interviewers. We aimed to limit Hawthorne effects–behaviour change in response to being observed–by conducting observations over a number of consecutive days in each selected facility and in a non-obtrusive, non-judgemental way, with our researchers making it clear to those being observed that all data collected would remain anonymous and that there would be no consequences of the observations for those being observed.

## Results

The QCN is a highly complex network in its composition, functioning and activities, operating at all levels of governance. In other papers from this collection, we analysed the complexity of the network's composition and functioning at global, national, sub-national and local levels [7,10,19–21]. In this paper, our analysis focuses on the effectiveness of the QCN through its various outputs and policy consequences as well as through the impact of such activities. We begin our analysis at the global level and then move to a cross-country analysis of QCN effectiveness at national and local levels in relation to the four LALA strategic aims–Leadership, Action, Learning, Accountability–envisioned by the QCN network as a whole. Table 1 summarises some of the key findings from our cross-country analysis.

### QCN effectiveness at the global level

Network activities at the global level, through the QCN secretariat led by WHO, focused on supporting countries with the LALA framework, developing standards and indicators for quality of care, providing technical assistance to MoHs and facilitating learning between countries. The QCN secretariat steered the formulation and distribution of standards, guidelines, and technical documents for quality of care (QoC). In particular, the WHO co-developed with global QCN partners (such as UNICEF, USAID, IHI, Jhpiego, URC ASSIST) and countries: the QoC standards for maternal and newborn care in 2016 –prior to the official launch of the QCN [22], the QoC standards for children and young adolescents in 2018 [23] and the QoC standards for small and sick newborns in 2020 [24]. As part of the network's monitoring strategy, the WHO co-developed 15 core quality indicators [25] in 2018, aligned with other global MNH initiatives [26–28], that were to be tracked in all QCN facilities and integrated in the countries' routine information systems to facilitate learning and accountability within and between QCN countries. Support and technical assistance were provided by the QCN secretariat to countries through regular calls, periodic implementation and technical briefs, field visits, and meetings with Ministries of Health (MoHs) to share implementation progress and challenges.

**Table 1. Key findings on QCN network effectiveness across case study countries.**

| | Bangladesh | Ethiopia | Malawi | Uganda |
|---|---|---|---|---|
| **Leadership** | | | | |
| Road map (operational plan) progress and awareness | 2017–2022 operational plan. 2019 Roadmap: standard operating procedures (SOP) for quality improvement 58% aware of operational plan, and 82% aware of SOP at sub-national and local levels | 2017–2020 roadmap developed in 2017, to be evaluated in 2022 34% of QCN health facility workers aware of roadmap | Roadmap available in 2017, fully developed by 2019 24% aware at sub-national and local levels | Developed 2018, revised 2020, operational plan late 2019. Low awareness at sub-national and local levels |
| **Action** | | | | |
| Learning sites and scale-up | UNICEF: five facilities in one district then 27 facilities in 6 districts. Save the Children: a few, then 56 facilities in one district. Scale-up to 298 facilities across 28 districts by early 2022 | 48 facilities across 14 districts Scale-up: not done by early 2022 | 25 facilities across 6 learning districts, not selected until 2019 Scale-up: not done by early 2022 | 18 facilities across 6 learning districts. Scale up: 3 additional regions covering 88 facilities |
| **Learning** | | | | |
| Integration of 15 core quality indicators into national health information systems | Government collects 41 indicators on quality of care including most of the 15 core quality indicators. Water and sanitation, and experience of care indicators remain unintegrated. | Parallel system to collect 15 core quality indicators, though integration planned. | Some indicators integrated in 2018. One experience of care indicator integrated, others remain unintegrated. | Some indicators integrated in 2019. Experience of care indicators remain unintegrated. |
| **Accountability** | | | | |
| Establishment of a mechanism for community engagement | Existing community involvement mechanisms pre-dating QCN, e.g. involvement of community leaders in monthly district leadership coordination meetings | Community elected client councils lead Community Score Card review process including health facility workers and service users | Included community engagement and social accountability as part of quality of care standards. Community score cards in some districts. Establishment of ombudsmen at facilities. Though ombudsman not independent, and not often used. | Included community engagement and social accountability as part of quality of care standards. Development of community dialogue guidelines for discussion of quality assessment results with communities. |

To further facilitate learning between countries, the QCN secretariat organised international meetings where all global partners and network countries sent delegations of eight to ten people, creating opportunities to evaluate progress of the network, share lessons learned and network between countries. These meetings were seen as facilitating both the Learning and Accountability facets of the LALA framework. Global and national participants noted that those meetings evolved in time as the network matured, from the global leadership and technical experts sharing information unidirectionally to national stakeholders, to countries sharing their progress and learning with one another whilst global actors took a background role. Participants also saw the international meetings as providing a necessary chance to *"take stock"*, evaluate, and challenge the progress of the network. Initially meant to occur annually, in practice only two international meetings have taken place: in February 2017 (Lilongwe), and then March 2019 (Addis Ababa); with smaller meetings focused on technical assistance occurring in between. Because of the COVID pandemic, no further international meetings took place since 2019, except a last meeting in March 2023 (Accra) to share final lessons from countries. In addition, the QCN secretariat launched a network website, shortly after the official launch of the network in 2017, which was used as a global learning platform to share resources on QoC best practices for professionals within and outside QCN [6]. The website consists of: a knowledge library with guidance documents for QoC implementation; a podcast to share

health practitioners QoC experiences; a global Community of Practice that any health practitioner can join to discuss ideas, practices and issues related to QoC for MNH; and monthly newsletters providing updates on the network's activities. Finally, the QCN secretariat facilitated regular webinars, open to anyone, on different topics and issues related to QoC where global partners present or where partner countries share updates, learning, and ideas (see [29] for more on the topic). Interviewees at the global level found this activity important to support and maintain momentum within the network. Overall, many global participants spoke of the impact of the learning emerging from the QCN, steered by the WHO secretariat, and how it should continue beyond the end of QCN as it requires little funding to maintain.

As demonstrated in the first paper of our collection [7], the QCN network at the global level, led by the efforts of the QCN secretariat, was effective in bringing together network countries' governments and global actors to raise the profile of QoC in MNH on the global agenda. The network at the global level was a catalyser for aligning goals between global actors and network countries, and supporting countries, leading to the emergence of the QCN network in-country.

## Leadership

As described in other papers from this collection [7,10], QCN was designed to give governments leadership of in-country implementation. The implementation approach therefore relied on MoHs to align the goals of in-country MNCH partners as well as resources to develop or strengthen national policies, strategies and structures for quality of care in [MNH] health services [8] in order to pursue the network's four strategic objectives. Thus, in-country, implementation was steered by the MoH through a technical working group (TWG) comprised of a variety of partners with the aim of developing or updating a national MNCH QoC roadmap and operational plan for the implementation of the QoC standards, as well as provide technical and financial support for QoC implementation. Funding for QoC implementation was also to be provided by MoH, according to countries' roadmaps [30–32], however government funding proved to be limited and unpredictable. In Ethiopia, thanks to their earlier work on QoC [7], by 2017 their TWG was formed and the national roadmap for 2017–2020 was developed and due to be evaluated in 2022. National key informants felt that the roadmap has clearly put goals and strategies for implementation including procedures, indicators and monitoring and evaluation. There was also regular meetings and communication among them to help cohesion. Yet, the roadmap was not well known amongst our survey participants working in QCN health facilities, as only 34% confirmed they were aware of it in Nov-Dec 2021. In Uganda, although the TWG was established in 2017, its work was relatively slow until 2019 when a champion was assigned, according to national interviewees. The roadmap was developed in 2018, then revised in 2020, with an operational plan devised end of 2019. Interviewees in our first round of data collection linked the lack of roadmap and direction early in the network's implementation with a lack of clarity and coordination, resulting in many actors undertaking various activities across the country without standardisation. Once implemented, our interview and survey results indicate a lack of awareness of the roadmap and operational plan among sub-national and local participants particularly, suggesting those might not be widely utilised. One reason could also be that at the frontline, these documents are presented in the form of tools or guidelines and not necessarily referred by the macro policy labels such as 'roadmap'. Additionally, interviewees have indicated that adhering to the roadmap and plan was challenging because of different factors but most especially the diversity and approaches of network partners who sometimes used additional tools, local targets or had different schedules. In Malawi, the TWG was established in 2017 and the roadmap made available in 2018. As the

roadmap and other strategic documents were not ready at the launch of the network, national interviewees in earlier rounds of data collection described a lack of strategic direction leading to delays in implementation and unorganised activities. In later rounds, national interviewees found the roadmap helpful by providing direction and outlining roles and responsibilities of partners. Some national implementation partners further believed the roadmap helped harmonise the efforts of various partners and ensured resources were used more effectively. In contrast however, awareness of the roadmap was still low (24%) among our 135 survey respondents in our last round in Malawi–predominantly from the sub-national and local levels–whilst awareness of other strategic QCN documents varied between 17% and 48%. Bangladesh developed a broad operational plan for health, nutrition and population for 2017–2022 which helped, along with additional national plans and guidance on quality of care (2015), patient safety (2018), and quality improvement (2019) to establish structures for quality improvement including those used in QCN. By md-2021, 58% of survey respondents at sub-national and national-level in Bangladesh were aware of the operational plan and 82% were aware of the quality improvement standard operating procedure.

The other component of the Leadership QCN strategic objective was the establishment of supportive governance structures in all countries to lead the QCN work within the health system. At the national level, all countries established a directorate within the MoH to oversee QI efforts, including for MNH, and hence QCN activities. Directorates were established before the inception of QCN, in Bangladesh, Ethiopia and Uganda, whilst Malawi set up theirs in 2016, ahead of the official launch of the QCN. The main functions of the national directorates are to oversee the coordination of QI activities and training in the country, establish quality standards, guidelines and standard operating procedures, and to monitor and evaluate them. Additionally, various national QI committees were established to support implementation in Bangladesh, although it has been difficult to assess their impact as few committee meetings took place over the years, according to our participants and our observations.

QCN governance was more developed at the national level of case study countries and lack of governance structures at sub-national and local levels impaired effectiveness of QCN activities. Where governance structures existed at sub-national or local levels their effectiveness was often impeded by a lack of capacity, lack of human resources or lack of ownership of responsibilities, and lack of support from national level.

*"Regional quality structure is very weak and it is understaffed. Even those assigned people have lots of responsibilities. They are assigned for every quality-related activity, not just for MNH—not only that—but also do other things and it is difficult"* (Implementing partner-National-Ethiopia Round 2)

Corruption was also mentioned as an issue, particularly in Uganda and Ethiopia. For example, in Uganda some key informants mentioned both cases of individuals diverting funds and systemic issues related to procurement of supplies and favouritism in relation to travel, training and mentorship opportunities.

Leadership at national level has been important to coordinate and monitor QI efforts in the countries. Funding for those functions however has been dependent on international donors, which impacted the QCN implementation in-country and raises issues for the sustainability of leadership efforts. Even in countries where network emergence was stronger [7] due to existing policies and initiatives and where leadership and QoC were more institutionalised, effectiveness of the network was still dependent on financial resources that are mostly external and on the commitment of governments to dedicate resources to QoC. In Ethiopia, the government has a budget allocated for quality improvement and QCN was part of it. This budget was small,

especially after funds were diverted due to the COVID-19 pandemic and to rehabilitate conflict-affected areas. Implementing partners supported the 48 facilities by supporting coaching and learning activities and supporting the MoH to prepare TWG and annual QCN meetings. In Bangladesh, efforts have been made to secure their own independent budget but that has not been achieved yet according to government participants.

## Action

The 8 WHO QoC domains and linked standards for maternal and newborn care [22] were adapted in 2017 in Ethiopia and Uganda, in 2018 in Bangladesh, and in 2019 in Malawi, with technical assistance from other QCN partners. Before this, there were no national QoC standards for MNH in Malawi, Uganda and Ethiopia, though quality of care was incorporated in national plans, while some facilities in Bangladesh had been using the Every Mother Every Newborn standards [33] since 2016. In all countries, the standards were used as a QI reference tool and to assess or audit health facilities, and in some countries, e.g. Uganda, different partners were focused on different standards. Many respondents thought the standards were useful to identify where improvements were needed, for example in Malawi:

"*the assessment and the standards are highlighting gaps in the system*" (Government-National-Malawi Round2)

Others thought more detail was required to improve quality of care, for example, in Bangladesh:

"*the QED* [QCN] *indicators are very generic indicators to me. It never reflects the quality improvement of a health facility. It should have more sub indicators.*" (Implementing Partner-National-Bangladesh Round 4)

Adaptation in Malawi and Uganda further included the addition of a ninth standard focusing on community engagement and social accountability in order to enhance community participation in improving QoC [30,31]. Interviewees at the national level pointed out that this additional standard was a necessary change for the success of the QCN. Regarding the other WHO standards developed by the QCN at the global level–those for children and young adolescents [23], and for small and sick newborns [24]–our data indicate that none of the countries have started to adopt them.

Under QCN's Action strategic objective, each country was to choose learning districts and health facilities, representatives of others in the country "to serve as laboratories for learning" for implementing QoC activities [9]. By keeping track of those activities and documenting the learning, it was expected that successful interventions would be scaled-up to other districts and facilities. Learning sites were selected in each country (Table 1) by the government, together with implementing partners, some of whom had pre-existing work in the selected districts. Selection of learning sites was based on maternal and neonatal death burden, geography, and pre-existing and planned work. In Ethiopia, the government allocated specific funding for QoC in those QCN facilities in addition to funding provided by implementing partners. Several regions were excluded due to security concerns given the political instability in Ethiopia at the time of selection. Scale-up was not achieved in Ethiopia or Malawi by 2022, whereas in Uganda, and especially Bangladesh some scale-up was achieved (Table 1). Implementing partners operated with a high level of autonomy and independence in terms of what activities to implement and how. Whilst the MoH tried to coordinate and track efforts, participants reported it not to be effective beyond *"big"* partners such as USAID. In Uganda, in 2019, the

MoH worked to improve coordination and strategic implementation by moving the ownership to a different department. Interviewees reported that, beyond the learning sites, implementing partners have been carrying out QCN activities in other districts they operate in. They further noted that this initial siloed and disjointed implementation attempt slowed down the success of the network in Uganda and that it was only three years into the five-year effort that a true network was beginning to form in the country. Although frontline health workers in Uganda were more aware of quality improvement efforts in general, led by the MoH and implementing partners, rather than QCN specifically. In Bangladesh, selection of sites differed as two initial pilot districts were selected in 2017 and implementation responsibilities were split between two partners: UNICEF and Save the Children (funded by USAID), who chose their own pilot learning facilities, either based on high mortality rate and remoteness or on lack of previous QI initiatives.

Whilst capitalising on existing work from partners to implement QoC interventions in QCN sites, this approach resulted in siloed working at sub-national and local levels. QCN activities in learning sites often differed in their content and scope between learning sites–depending on the implementing partners' presence in the district–and their implementation was dependent on the available partners' resources and technical support, as financial support to learning sites from the government was limited or non-existent.

*"But I find the issue to do with financing more of a cause for us to fail. This is because I look at all the components of the health system and I find. . . well. . . I was trying at this particular time to think about the investments that have happened for example in [Case 1-low performing], as a learning district. How much did government commit to the goal that we reduce the maternal mortality rate by 50% in the implementing [of QCN] in the nation and districts by 2022? If we are to be honest, success of every implementing district was dependant on the kind of and the flexibility of partners that are in the district"* (Implementing partner-National-Malawi Round 3)

This was further noticed in our health facility observations, where these fluctuations presented a challenge to track what QoC activities on site were considered QCN outputs. In case study QCN sites in Bangladesh, QI activities were more easily identifiable as they were part of the determined bundle of interventions implemented by either Save the Children's or UNICEF's projects (depending on the case study). In most case study QCN sites in Ethiopia, Malawi and Uganda, it proved more difficult as many health facility workers were not aware of QCN (Fig 1) or were not able to attribute if a QoC activity was a QCN output. Generally, the type of QCN activities observed in case study sites included: review of routinely collected data, assessment of facilities against QoC standards, Maternal and Perinatal Death Surveillance and Response, community engagement, point of care QI across all of MNH continuum of care. Whilst many respondents at local levels were not aware of QCN, they were aware of quality improvement efforts more broadly, including those pre-dating QCN.

Overall engagement with QCN was perceived to be higher (and rising) in Uganda, and Bangladesh (though falling) than in Malawi or Ethiopia (Fig 2). In Bangladesh a median of 10–20 hours over 6 months was reported to have been spent directly on network activities, in Malawi 5–10 hours, Ethiopia 5–10 hours, and in Uganda 5–10 hours rising to 20–30 hours per 6 months (Fig 2). A greater proportion of respondents in Bangladesh and Uganda also reported that their views and ideas contributed to the network and that they were able to drive the network agenda (Fig 2).

As per the LALA framework, network outputs at sub-national and local levels further involved a capacity building element for QI interventions and monitoring QoC such as QoC training, on-site QI coaches and QI committees within the health facilities. Capacity building

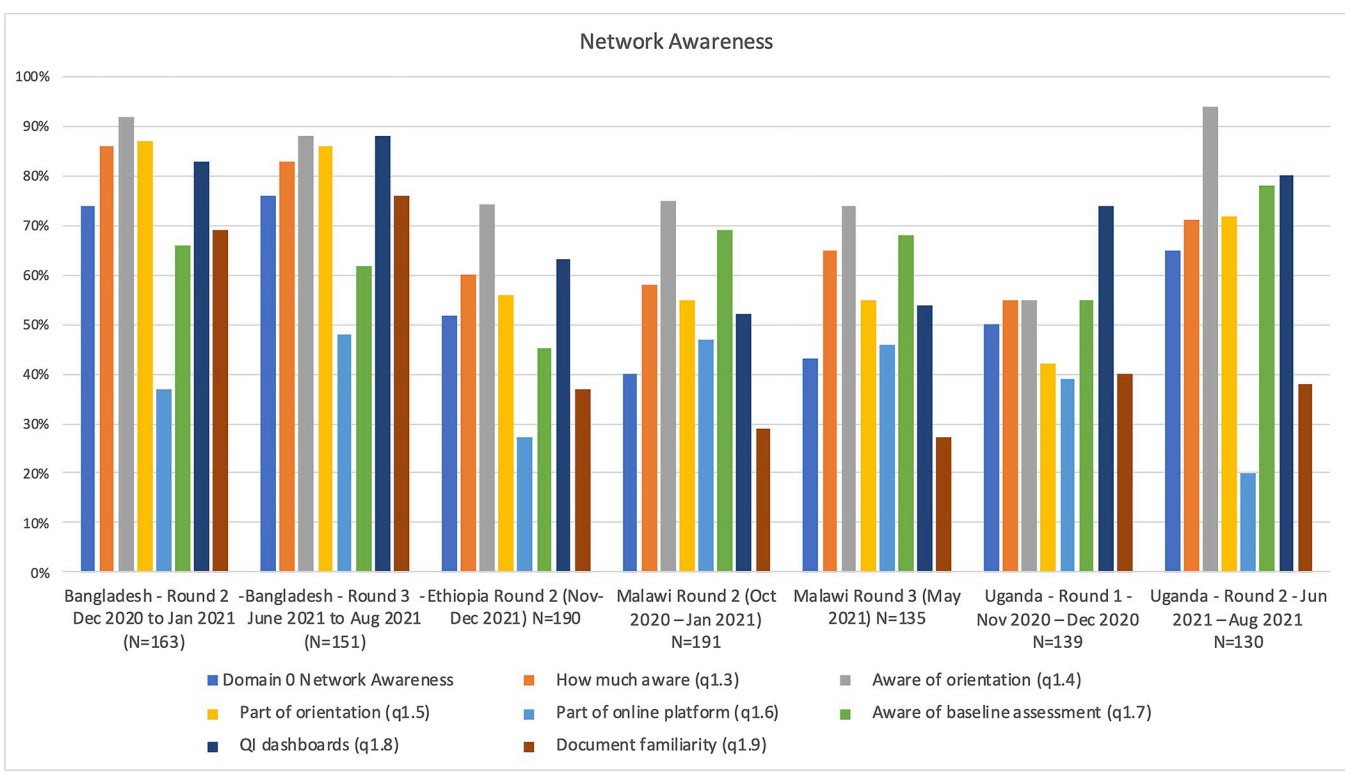

**Fig 1. Network awareness (survey data).**

and coaching visits covered a wide range of topics, such as intervention packages, data collection and monitoring, as well as training QI committees in developing, implementing and monitoring QoC activities. QI committees were the QCN output most discussed in interviews and observed in case study sites, corroborated by our survey respondents at health facility level whose majority indicated they were part of a QCN/QI committee. Indeed, the QCN led to the establishment or development of various QI committees in health facilities in Bangladesh, Ethiopia, Uganda and Malawi. Including mostly frontline healthcare workers, those committees met regularly (except during COVID-19 disruptions) to identify and prioritise QoC issues in the facility. The resulting local improvement projects were partially successful in some locations (e.g. reducing the incidence of birth asphyxia in a facility in Malawi), sometimes via logistical and funding support of implementing partners (this was mentioned in all countries), and unsuccessful in other locations (e.g. not reducing neonatal deaths in a hospital in Malawi).

In Bangladesh, Malawi and Uganda, capacity building outputs were mainly conducted by implementing partners (with collaboration on some occasions with local academic partners in Bangladesh). In Uganda some respondents thought not optimizing having Makerere University School of Public Health as the research and learning partner was a missed opportunity. In Ethiopia, despite being government-led, capacity building was supported financially and technically by implementing and technical partners. Therefore, consistency of capacity building activities throughout the implementation period was dependent on partners' presence and resources for learning sites, leading to variable levels of activity between sites.

*"It's much more related to financial muscles and if the committee had a proper funding to run its activities, like its plans to do its supervision at whatever time they should be able do their*

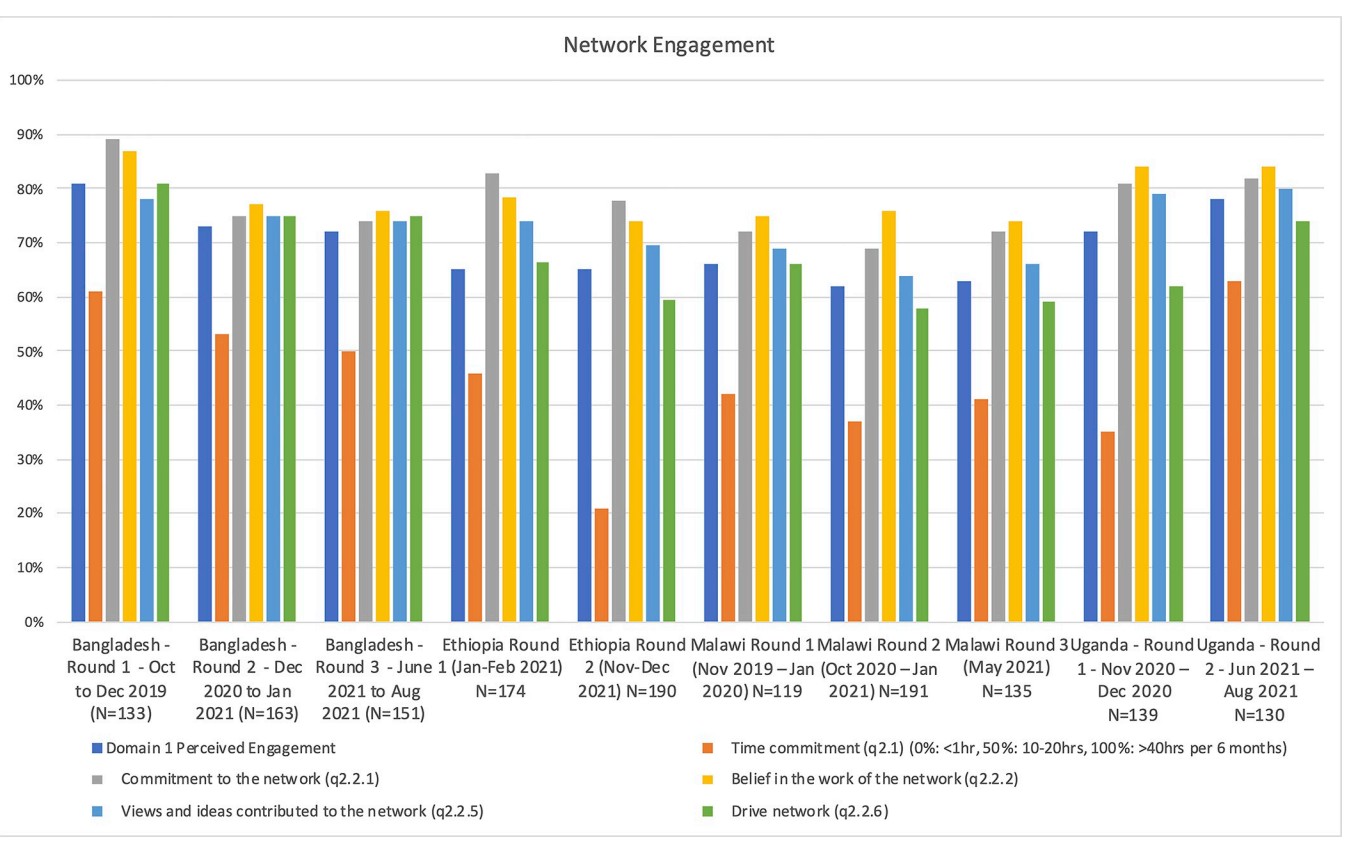

**Fig 2. Network engagement (survey data).**

*activity. Unlike now where they have to wait for partners to come in and help. [. . .] So, if there was a proper financial support it would be good to the network"* (Government-Local Case 3-Malawi Round 1)

Learning sites activities, and their scale-up, were dependent on the presence and resources of one or several implementing partners, meaning that the amount and duration of resources provided by the implementing partner(s) dictated the level of effectiveness of the network at sub-national and local levels.

*"When you bring a five-year learning project, it should have its own resources for implementation and to scale it up with the lessons learnt. This is the reason why the footsteps of the partners were followed. It doesn't have its own resource. Partners have their own interests; they all have different approaches that they follow. For example, we say learning collaborative should be prepared in three months. Some do it within six months. Some conduct coaching every month, the others do it quarterly. Therefore, it lacks uniformity. This is still because there is no resource. So, partners plan with their own resources. Due to these problems, I can't say that things have gone the way we want them to."* (Government-National-Ethiopia-Round 1)

For instance, in Bangladesh, committed partners for the duration of QCN implementation, like Save the Children and UNICEF, meant that a constant level of activity was maintained in the QCN facilities they operated in with some change to show for. They were, however, themselves funded by global partners and therefore also dependent on volatile external funding. For

example, when UNICEF funding was reduced in some districts this meant that QCN activities carried out by UNICEF staff had to be taken over by health facility workers leading to a drop in QCN activities. On the other hand, some learning sites (e.g. in Malawi and Uganda) had no or little partners involved, leading to a dearth of QCN activities in those sites. For example, as reported by a local stakeholder in Malawi:

*". . .we have so many projects that we will put on paper and the issues of training QIST members so that they should function properly, we should know their roles and responsibilities. Now, we would have loved if we had a potential funder or sponsor of this program so that at least we know that when we have activities, these are the ones that are going to support us. But as it is now, there is no donor who has come openly to say that we are going to support this program. So, it's a challenge to us"* (Government-Local Case 3-Malawi Round1).

In Uganda, there was some attempt to pool resources from partners, but this effort failed because of lack of trust, and a lack of harmonisation of different tools and methods used by different partners–both exacerbated by the leadership vacuum at the start of QCN in Uganda which later improved. Additionally, as observed in several facilities, this has meant that when partners' projects came to an end in a given area due to financial constraints or end of funding, partners ceased to support the learning sites–making it difficult for sites to continue QCN activities given their lack of human, technical and material resources, as well as undermining M&E efforts.

*"When IHI, WHO, CHAI pulled-out from the network, it was very difficult to cover the sites which were under their organizations. Taking lessons from this, we are working to make the government takeover the activities. There were gaps at collection of common core indicators when the partners pull-out. Partners always depend on the donors as a result there are challenges."* (Government-National-Ethiopia Round 2).

The COVID-19 pandemic further contributed to weakening partners' support to activities in learning sites [21]. Issues around resources dedicated to QCN are taking place in a context of weak health system capacity, in terms of material and human resources, undermining network effectiveness in all countries. This was the challenge to QCN activities most reported by participants in all countries. The lack of human (including support staff) and material resources was also observed in our case studies facilities as a hindering factor of implementation.

*"Looking at the 8 standards for MNCH that we adopted, we realize that implementing those standards isn't just about quality improvement. We need real inputs and we have seen that without improving infrastructure, without having some of the services or the equipment, for example the newborn care to improve it, we need to have actual equipment, actual services, people need to have skills for managing those newborns. Some of the inputs were not well catered for even in the 6 learning districts and probably that's why we did not see any reduction in the newborn mortality. So, we have seen that the input level as we implement it, the inputs that we plug into probably were not well addressed. Even just the technical skills not just the quality improvement skills but the technical skills for providing services needed to be strengthened from time to time and needed to be in contact with the people implementing them to be supported at least on a quarterly basis. This did not happen for most of the districts, and so we know why we did not perform well."* (Government-National-Uganda Round 2)

## Learning & accountability

Learning and Accountability are two important strategic objectives of the QCN. Here we discuss the infrastructures put in place in the four countries to support learning and accountability. We analysed elsewhere the diffusion of information, innovation and learning processes within the network [29]. Part of this strategy, as mentioned earlier, was the development in 2018 by WHO, global partners and network countries of 15 core quality indicators for MNH–an immense effort when considering the situation before QCN. Among global interviewees, there was general agreement that the dedicated working group had done good work and the indicators were well-received, yet by 2019 participants still described a lack of clarity on how and when data should be collected. The aim was to integrate those core quality indicators to the countries' routine Health Information Systems (HIS), which required according to some global and national interviewees, a lot of technical support from the network secretariat (with limited manpower) and partners. Despite the complexity of the process, each country has had some success in integrating the 15 core quality indicators into their national health information systems (Table 1), though experience of care indicators have been a challenge, with independent regular systems of data collection from patients proving difficult. Data in learning sites were often collected by implementing partners. In Bangladesh, both the government and the major implementing partners aimed to place quality improvement indicators on the web-based national dashboard, which can be used to download custom reports on various indicators. National interviewees indicated that this dashboard was largely functional, and stakeholders expressed pride in their ability to roll out this new initiative while simultaneously responding to COVID-19. In Ethiopia on the other hand, the network relied on a parallel system to their HIS to collect data on the QCN quality indicators (data from facilities reported up in the system electronically via spreadsheets provided by MoH) until a planned integration to DHIS2 at the next HIS revision. This proved to be a challenge according to our interviewees who reported that the QCN indicators did not align with the previous reporting system and questioned the robustness of the experience of care indicators since the data was not collected by independent actors not involved in the care provided. However, all countries faced structural issues and a lack of capacity in collecting good quality data. For instance, countries still relied on separate surveys to collect data on experience of care indicators, unlike indicators on provision of care that were captured by DHIS2 or a parallel system (for Ethiopia). Additionally, in all countries, capacity building on data collection, analysis, quality and management was crucial due to the lack of capacity at sub-national and local levels but depended on the efforts of the partners supporting the learning facility. Supported by our facility observations, several participants in all countries further brought up concerns over the reliability of the data reported, particularly around patient experiences and mortality figures. The work of the QCN on indicators and monitoring did bring attention to the importance of health data for QoC improvements. In Ethiopia for example, national and local interviewees considered that the network had improved health data documentation, management, use and reporting in health facilities, even if gaps in capacity remain. In our survey, most respondents in Bangladesh (83% rising to 88%) and Uganda (74% rising to 80%) indicated that there were quality improvement indicator dashboards or visualisations at their facility (Fig 1). This was 63% in Ethiopia and just over 50% in Malawi. Locally, in our observations in health facilities, behaviour change around data monitoring seemed to vary. Some of the sites had a system of bulletin boards which visually displayed QI metrics for the preceding several months, including maternal and neonatal mortality, as a means of making the information more accessible and of motivating staff. Some facilities, for example those observed in Bangladesh, also regularly completed partographs to monitor the progress of labour as part of efforts to reduce mortality. In some

facilities, data was also discussed in the facility QI committee to improve accuracy and enhance accountability. A few facilities were observed to not be using dashboards, visualising data or using monitoring indicators in their practice. At the end of our data collection, there was still a lot of work to achieve in all countries to collect (reliable) data, integrate to DHIS2, increase the quality of the data collected and improve analysis at the national level. However, from our interviews and observations, the lack of human resources and capacity around M&E makes this difficult to achieve.

> *"Yes. . . I think we have not done well in the M&E. I think there is a lot that we would have done. Maybe the problem is that some of the problems are beyond us, because we need full time M&E, looking at the data and ensuring that reports are being done. And when reports are put in the system, analysing the information or extracting the information from whatever platform is there and pushing it back to policy makers and the facilities, so that they can use that information to further improve the network. So, there is a lot that needs to be done in the M&E side"* (Government-National-Malawi Round 2)

As a result, the data received from countries was often incomplete or of poor quality leading to a lack of quantitative analysis on how well the network has done on key process and outcome indicators and whether the overall goal of the network of reducing case fatalities by 50% has been achieved. Therefore, the impact of the network on reducing mortality remains unknown, though may be reported in 2023 (WHO-Global Interviews-Round 2). For global interviewees, this goal was always *'ambitious'* or *'aspirational'* and they were cautious about the network's ability to achieve it within the five-year timeframe. Among global actors there was an overall sense that the network needed these bold, ambitious targets in order to gain momentum and attract engagement, funding, and global attention. Most global actors believed these goals were an essential catalyst at the start of the network and continued to serve as important motivation. Some national and local participants in Bangladesh, Ethiopia, Malawi and Uganda believed that some progress had been achieved towards reducing maternal and/or neonatal deaths in some of the learning sites. Our survey results indicated that most respondents in each country perceived the network to be valuable, though more so in Uganda and Bangladesh than in Ethiopia or Malawi (Fig 3). Fig 3 also shows a breakdown of indicators within this domain, with the highest scores being for "would recommend joining the network", and the network "helps me professionally", and lower scores for perceiving QCN to have resulted in healthcare improvements.

As part of the Learning and Accountability strategy, countries were also to involve a research institution to document the lessons learned from the network in order to facilitate the scale-up of the learning to non-learning sites. In Bangladesh and Uganda, the MoH built upon existing partnerships with academic partners such as NIPSOM and Makerere University School of Public Health respectively to achieve this milestone. In Uganda, Makerere was not optimally engaged as the learning partner, working instead in more of a consulting and advisory role for the MoH and WHO, with limited funding, and did not get to interact widely and in-depth with network members. At the end of our data collection, this milestone had not been achieved in Malawi and Ethiopia. From our qualitative data, despite the learning achieved in-country across different platforms [29], the network was not mature enough in any country to facilitate the scale-up of the learning from laboratories to non-QCN facilities through the national research institution in a systematic way. This was due to a combination of factors including lack of funding, an over-reliance on implementing partners and delays due to the COVID-19 pandemic. In Bangladesh, at the time of our last data collection in 2022, NIPSOM had advanced plans to scale-up QoC training in all districts, via allocated funding from the government budget.

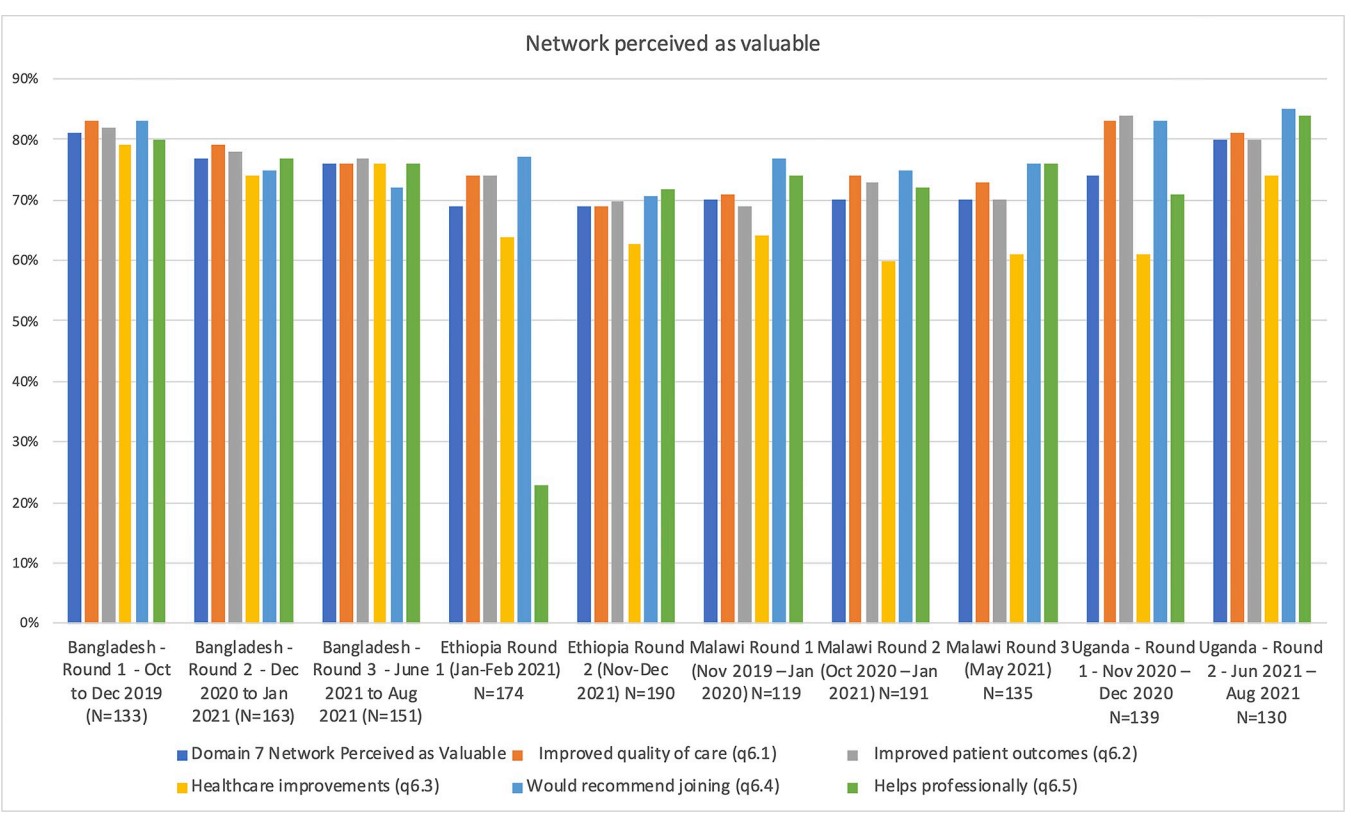

**Fig 3. Network Perceived as Valuable (survey data).**

Lastly, the Learning and Accountability strategy involved the establishment of a mechanism for community engagement in QoC in the learning sites, in line with the second global goal of the QCN to improve experiences of care for women and their families. Each country took a different approach to include community voices in the learning sites to improve QoC and accountability (Table 1). Some of these built on earlier work with communities pre-dating QCN, though were not always fully integrated with QCN [34]. Some community discussions and engagement mechanisms were developed into action plans with learning facilities, although in practice our data indicate that those are not fully optimised. All in all, the global QCN goal to improve experiences of care and "*enable measurable improvements in user satisfaction with the care received*" [8] seem to have been relatively neglected in national QCN implementation in all countries. Mirroring this at the global level, documents collected show an emphasis on community engagement in the early phases of QCN development (especially pre-launch) but overtime, little or no mention of this component is made. Malawi achieved the most regarding this goal. Indeed, Malawi was the only country that integrated one indicator (percentage of facilities with functional ombudsman and suggestion box or other feedback mechanism) to DHIS2 linked to patient experience of care. Malawi's establishment of an ombudsman structure was criticised though for not being independent and consequently rarely used.

## Effect of COVID-19, and political instability

The COVID-19 pandemic impacted network effectiveness related to the Action, Learning and Accountability strategic objectives as those were not as advanced as the Leadership one by the

time the pandemic hit. Indeed, many participants at all levels of governance mentioned that QCN resources were diverted towards Covid-19; some learning facilities were repurposed to only accommodate Covid-19 patients; focus remained on Infection Prevention Control over all other aspects of QoC; and local restrictions kept patients away. Consequently, participants believed this led to important delays in implementation of interventions and scale-up as well as learning and accountability outputs. Local respondents in all four of our case study countries further indicated that all QCN activities in their facility ceased during periods of heightened restrictions. Additionally, all QCN meetings, trainings and interactions at different levels of governance reduced in frequency, even when an online option was available. Some of the frontline workers interviewed further believed that progress towards reducing case fatalities rates was stalled as a result of the pandemic. Additionally, some of the countries faced periods of instability around elections (e.g. Malawi and Uganda) and conflicts (e.g. Ethiopia) that impacted the work of the network [21].

## Discussion

Overall, we found the 'leadership' strategic objective of QCN to be most advanced, particularly at global and national levels, though efforts to improve co-ordination, integrate and harmonise work are on-going. QCN built on long-standing commitments and initiatives to improve maternal, newborn and child health and much progress has been made toward the 'action' and 'learning and accountability' strategic objectives of QCN over the 2017–2022 period. Many gaps remain though, including those related to capacity building and mentorship, community engagement, and collection, quality and use of data on experiences of care. National scale-up has not yet happened in any of our case study countries, though may do in Bangladesh soon. The gradual pace of QCN implementation was due to a combination of factors including an over-reliance on implementing partners and donors, lack of earmarked government funding, and the COVID-19 pandemic. As several of our global level respondents also suggested, it may be that an initiative like QCN requires several more years to become fully embedded in government health systems and fully operational to the point where significant impacts on maternal, newborn and child case fatality rates may be achieved (and measurable). The level of coordination, collaboration and complexity involved requires time to work through and long-term budgets, as illustrated by the guidance on district implementation developed by QCN only being published in September 2022 [35], after the end of our data collection. Going forward, scale-up and implementation efforts to improve quality of care needs to be done with dedicated budget lines. Ideally, domestic funding would also be directed towards homegrown quality improvement expertise.

Leadership of the network in each country by the MoH is a strength of QCN and a positive step toward achievement of QCN strategic objectives. It reflects national commitments to improve quality of care, following earlier focus on increasing coverage of services (S3 Text). The 11 'pathfinder' countries of QCN were selected based on national buy-in to the quality-of-care agenda. Strong leadership and buy-in may not have happened without the support of global partners though. Dependence on implementing partners and donors often meant MoHs had a lack of real control or ability to integrate, harmonise and scale-up the separate, complex, and often pre-existing, implementation efforts QCN started in different sub-national or local areas in each country [10]. This was also compounded by a lack of organisational and policy capacity, especially in Malawi, and external shocks such as the COVID-19 pandemic in all countries, and the conflict in Ethiopia [21]. We found the emergence of QCN to be greatest in Bangladesh [7] and this is reflected in Bangladesh being closest to national scale-up in this paper.

A systematic review of clinical networks published in 2016 and only identifying studies from high-income countries for inclusion [36] found quantitative evidence of effectiveness to

be lacking. The qualitative studies summarised in this review found that networks with a positive impact on quality of care and patient outcomes had sufficient resources, effective communication and collaboration, efficient management, and credible leadership [36]. Our work corroborates these findings, and in this paper, and our papers on QCN emergence [7] and legitimacy [10], we provide an in-depth analysis specific to QCN. The point on resources available for QCN activities is pertinent and comes into focus when considering that the amounts made available domestically were less than those from external partners (i.e., many activities were contingent on donor funding), and the external partner contributions were small per country per year in comparison to total spending on maternal, newborn and child health. Networks are also more likely to be effective if there is involvement of local actors in decision-making [11] and there are pre-existing relationships between actors–which was the case here. But a varied and inclusive composition of the network did not necessarily translate into network cohesion. The local peripheral levels of the network in each country were often insufficiently involved, such that the network was found to be far stronger and tangible at national and global levels than at local level [10]. In our accompanying stakeholder network analysis, we also found few local-to-local level links between QCN members in each of our case study countries–QCN was manifest as a 'hub and spoke' network in each country, with a low density of connections between actors, rather than a mature highly-linked network [20]. Our analysis of the sustainability of QCN [34] found that, due to lack of resources and time to embed innovations at local levels, QCN may not be sustained in its original form. Efforts to institutionalize QCN innovations in existing systems could mean aspects of QCN are carried forward within broader government quality improvement initiatives though [34].

It was widely agreed that the 2017–2022 timeframe of QCN was relatively short, and the ambitious "50% case fatality rate reduction" goal was more of a rallying cry than something QCN stakeholders thought feasible to achieve (and measure) in just five years. Given this, even though much remains to be done, what QCN has achieved so far, especially given the disruptions of the pandemic, can be considered a success. However, meso and macro systems improvements can be much harder to achieve than frontline improvements at learning sites, and this has not been a big focus of the network so far. They are often needed to enable frontline improvements, for example, to improve availability of well trained and motivated staff, ensure timely supplies of drugs and equipment. Embedding quality throughout the system is a longer-term project–so ideally this network needs to run for at least another 5 years to achieve strong results on quality and case fatality rates nationally in each country.

The key strengths of our work are the iterative nature of our inquiry over multiple rounds of data collection, in four countries and at the global level of QCN, over a three year period (2019–2022), and our use of multiple methods of data collection (interviews, observations, survey, document review), involving a diverse variety of QCN stakeholders, and our subsequent integrated analytical synthesis of our data, with reference to other studies we have undertaken concurrently as part of our wider QCN evaluation (S1 Text). Prior work of this kind has focused on advocacy and agenda-setting networks. Our work is also unique in focusing on an implementation network. The main limitations of our research are not interviewing or surveying service users, and not being able to quantitatively evaluate the effectiveness of QCN. Both were beyond the scope of our research due to limits on our funding and time, and lack of availability of required QCN quantitative data.

## Conclusion

QCN built on varied but visible foundations to further align efforts by ministries of health, implementing partners and donors, to tackle health facility-based maternal, neonatal and child

mortality and morbidity. It has had some success so far though there are many steps still to take to embed, improve, integrate, scale-up and sustain the nascent work started in health facilities in QCN countries. Continued ambition, commitment, and long-term, ideally domesticated, funding is required to continue the journey QCN has started.

## Supporting information

**S1 Text. PLOS GLOBAL HEALTH QCN Evaluation Collection 2-page summary.**
(DOCX)

**S2 Text. QCN papers common methods section.**
(DOCX)

**S3 Text. QCN papers common country context.**
(DOCX)

## Acknowledgments

We thank all respondents and stakeholders for their time and contributions toward making this work possible. The QCN Evaluation Group is: Nehla Djellouli, Kasonde Mwaba, Callie Daniels-Howell, Tim Colbourn (UCL Institute for Global Health, UK), Kohenour Akter, Fatama Khatun, Mithun Sarker, Abdul Kuddus, Kishwar Azad (BADAS-PCP Bangladesh), Kondwani Mwandira, Albert Dube, Gladson Monjeza, Rachel Magaleta, Zabvuta Moffolo, Charles Makwenda (Parent and Child Health Initiative, Malawi), Mary Kinney, Fidele Mukinda (independent researchers, South Africa), Mike English (Oxford University), Yusra Shawar, Will Payne, Jeremy Shiffman (Johns Hopkins University, USA), Kathy Lubowa, Agnes Kyamulabi, Hilda Namakula, Gloria Seruwagi (Makerere University, Uganda), Anene Tesfa, Asebe Amenu, Theodros Getachew, Geremew Gonfa (Ethiopia Public Health Institute, Ethiopia), Seble Abreham, Tanya Marchant (LSHTM, UK)

## Author Contributions

**Conceptualization:** Nehla Djellouli, Yusra Ribhi Shawar, Gloria Seruwagi, Jeremy Shiffman, Tim Colbourn.

**Data curation:** Nehla Djellouli, Kasonde Mwaba, Kohenour Akter, Gloria Seruwagi, Tim Colbourn.

**Formal analysis:** Nehla Djellouli, Yusra Ribhi Shawar, Kasonde Mwaba, Kohenour Akter, Gloria Seruwagi, Asebe Amenu Tufa, Geremew Gonfa, Kondwani Mwandira, Agnes Kyamulabi, Tim Colbourn.

**Funding acquisition:** Yusra Ribhi Shawar, Gloria Seruwagi, Jeremy Shiffman, Mike English, Tim Colbourn.

**Investigation:** Nehla Djellouli, Yusra Ribhi Shawar, Kasonde Mwaba, Kohenour Akter, Gloria Seruwagi, Asebe Amenu Tufa, Geremew Gonfa, Kondwani Mwandira, Agnes Kyamulabi, Jeremy Shiffman, Mike English, Tim Colbourn.

**Methodology:** Nehla Djellouli, Yusra Ribhi Shawar, Kasonde Mwaba, Kohenour Akter, Gloria Seruwagi, Asebe Amenu Tufa, Geremew Gonfa, Kondwani Mwandira, Agnes Kyamulabi, Jeremy Shiffman, Mike English, Tim Colbourn.

**Project administration:** Nehla Djellouli, Kasonde Mwaba, Kohenour Akter, Gloria Seruwagi, Asebe Amenu Tufa, Geremew Gonfa, Kondwani Mwandira, Tim Colbourn.

**Resources:** Nehla Djellouli, Kasonde Mwaba, Tim Colbourn.

**Software:** Nehla Djellouli, Kasonde Mwaba, Tim Colbourn.

**Supervision:** Nehla Djellouli, Yusra Ribhi Shawar, Kasonde Mwaba, Jeremy Shiffman, Mike English, Tim Colbourn.

**Validation:** Nehla Djellouli, Yusra Ribhi Shawar, Kasonde Mwaba, Kohenour Akter, Gloria Seruwagi, Asebe Amenu Tufa, Geremew Gonfa, Kondwani Mwandira, Agnes Kyamulabi, Jeremy Shiffman, Mike English, Tim Colbourn.

**Visualization:** Nehla Djellouli, Tim Colbourn.

**Writing – original draft:** Nehla Djellouli, Tim Colbourn.

**Writing – review & editing:** Nehla Djellouli, Yusra Ribhi Shawar, Kasonde Mwaba, Kohenour Akter, Gloria Seruwagi, Asebe Amenu Tufa, Geremew Gonfa, Kondwani Mwandira, Agnes Kyamulabi, Jeremy Shiffman, Mike English, Tim Colbourn.

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
