## [Decision Letter · Decision Letter 0]

3 Jul 2023

PGPH-D-23-00351

Effectiveness of a multi-country implementation-focused network on quality of care: delivery of interventions and processes for improved maternal, newborn and child health outcomes

Dear Dr. Colbourn,

Thank you for submitting your manuscript to PLOS Global Public Health. After careful consideration, we feel that it has merit but does not fully meet PLOS Global Public Health’s publication criteria as it currently stands. Therefore, we invite you to submit a revised version of the manuscript that addresses the points raised during the review process.

Please note that we have only been able to secure a single reviewer to assess your manuscript. We are issuing a decision on your manuscript at this point to prevent further delays in the evaluation of your manuscript. Please be aware that the editor who handles your revised manuscript might find it necessary to invite additional reviewers to assess this work once the revised manuscript is submitted. However, we will aim to proceed on the basis of this single review if possible. 

We look forward to receiving your revised manuscript.

Kind regards,

Jianhong Zhou

Staff Editor

Journal Requirements:

1. Please provide additional details regarding participant consent. In the ethics statement in the Methods and online submission information, please ensure that you have specified what type you obtained (for instance, written or verbal, and if verbal, how it was documented and witnessed). If your study included minors, state whether you obtained consent from parents or guardians. If the need for consent was waived by the ethics committee, please include this information.

2. We noticed that you used "unpublished data" in the manuscript. We do not allow these references, as the PLOS data access policy requires that all data be either published with the manuscript or made available in a publicly accessible database. Please amend the supplementary material to include the referenced data or remove the references.

Additional Editor Comments (if provided):

Reviewers' comments:

Reviewer's Responses to Questions

**Comments to the Author**

1. Does this manuscript meet PLOS Global Public Health’s publication criteria? Is the manuscript technically sound, and do the data support the conclusions? The manuscript must describe methodologically and ethically rigorous research with conclusions that are appropriately drawn based on the data presented.

Reviewer #1: Yes

2. Has the statistical analysis been performed appropriately and rigorously?

Reviewer #1: Yes

3. Have the authors made all data underlying the findings in their manuscript fully available (please refer to the Data Availability Statement at the start of the manuscript PDF file)?

Reviewer #1: No

4. Is the manuscript presented in an intelligible fashion and written in standard English?

Reviewer #1: Yes

5. Review Comments to the Author

Reviewer #1: The manuscript addresses an important component of Quality of care, an aspect to focus towards reducing Maternal, Newborn and Child deaths. Few comments;

1. Justification of choosing the 4 countries out of 11 need to be clear. Narration of the 4 selected counties have been provided but NOT clear why on these four countries only.

2. The effectiveness: Intervention delivery and process could be affected by the Global set up at the National level withing countries but also the involvement of international partners and other donors with parallel projects in these countries. This could have affected the country ownership and regarded as another project to be learned. This came out very clearly in all the findings of the four countries i.e Low awareness especially at the Sub national level (Meso and Micro levels), Delay or No integration, an extreme example from Ethiopia using parallel system, low implementation at the local level in all 4 countries except for Bangladesh where there was already an existing platform from Nutrition and population, with low accountability. In most of these countries, the learning came from international partner driven efforts like UNICEF, Save the Children and USAID with minimal efforts to call for country partnership, leveraging resources and create Government commitment to allocate specific budget lines for implementation at Sub national levels. YET, the evaluation has not shared any learning from the Global level.

The authors may need to comment on this, so we learn on Global responsibilities in supporting these initiatives and bringing together partners within countries including the National and sub national level. Contextual factors need to be thought and discussed.

3. The authors highlighted that the QCN was siloed and disjointed but didn't recommend strongly on what need to be considered as we set up these initiatives at National and local level as well as Global.

4. Data from Global view is critical, not sure why excluded in the survey

5. Ownership at the country level - require explanation

6. PLOS authors have the option to publish the peer review history of their article (what does this mean?). If published, this will include your full peer review and any attached files.

**Do you want your identity to be public for this peer review?** For information about this choice, including consent withdrawal, please see our Privacy Policy.

Reviewer #1: **Yes: **NAHYA SALIM MASOUD

---

## [Decision Letter · Decision Letter 1]

5 Oct 2023

PGPH-D-23-00351R1

Effectiveness of a multi-country implementation-focused network on quality of care: delivery of interventions and processes for improved maternal, newborn and child health outcomes

Dear Dr. Colbourn,

Thank you for submitting your manuscript to PLOS Global Public Health. After careful consideration, we feel that it has merit but does not fully meet PLOS Global Public Health’s publication criteria as it currently stands. Therefore, we invite you to submit a revised version of the manuscript that addresses the points raised during the review process.

We look forward to receiving your revised manuscript.

Kind regards,

Anteneh Asefa Mekonnen, Ph.D., MPH

Academic Editor

Journal Requirements:

2. We noticed that you used "unpublished" in the manuscript. We do not allow these references, as the PLOS data access policy requires that all data be either published with the manuscript or made available in a publicly accessible database. Please amend the supplementary material to include the referenced data or remove the references.

Additional Editor Comments (if provided):

Reviewers' comments:

Reviewer's Responses to Questions

**Comments to the Author**

1. If the authors have adequately addressed your comments raised in a previous round of review and you feel that this manuscript is now acceptable for publication, you may indicate that here to bypass the “Comments to the Author” section, enter your conflict of interest statement in the “Confidential to Editor” section, and submit your "Accept" recommendation.

Reviewer #1: All comments have been addressed

Reviewer #2: All comments have been addressed

2. Does this manuscript meet PLOS Global Public Health’s publication criteria? Is the manuscript technically sound, and do the data support the conclusions? The manuscript must describe methodologically and ethically rigorous research with conclusions that are appropriately drawn based on the data presented.

Reviewer #1: Yes

Reviewer #2: Yes

3. Has the statistical analysis been performed appropriately and rigorously?

Reviewer #1: Yes

Reviewer #2: Yes

4. Have the authors made all data underlying the findings in their manuscript fully available (please refer to the Data Availability Statement at the start of the manuscript PDF file)?

Reviewer #1: Yes

Reviewer #2: Yes

5. Is the manuscript presented in an intelligible fashion and written in standard English?

Reviewer #1: Yes

Reviewer #2: Yes

6. Review Comments to the Author

Reviewer #1: The authors has addressed all comments as per reviewers recommendations. This is an important information to be shared to the public.

Reviewer #2: Thank you for the opportunity to review this revised manuscript (I was not part of the previous round of review), which provides a fascinating account of the work done by a network (QCN) launched in Malawi in 2017. This network embodies the values of ethical global health principles as part of its strategic objectives. This paper is the third account of evaluation of the work done in four of the 11 GCN countries. The study design is that of multiple embedded case studies, reviewing perspectives and data sources across a 6-year period (2016 - 2022). A staggering number of 248 interviews were conducted across a 3 year period, involving key stakeholders across all levels of governance. The authors provided an account of measures implemented to navigate the challenges posed by the Covid pandemic.

The following aspects could enhance the manuscript further, mainly to ensure complete reporting in line with the COREQ guidelines:

1. Please clarify the background and qualifications of the members of the QCM Evaluation Group who conducted the interviews. What was their relationship with the context and participants?

2. Please clarify the process of "non-participant" observations: how did these members of the Evaluation Group prevent possible Hawthorne effects?

3. It sounds like this was a large group of members in the Evaluation Group - how did you ensure consistency in the fieldwork and minimize any potential impact on validity and trustworthiness? How would you account for the statements of positionality/reflexivity in terms of the relationships with the contexts and participants? It is reassuring to read the considerations around team meetings of coders and how the results of the survey was used to triangulate with the qualitative data.

7. PLOS authors have the option to publish the peer review history of their article (what does this mean?). If published, this will include your full peer review and any attached files.

**Do you want your identity to be public for this peer review?** For information about this choice, including consent withdrawal, please see our Privacy Policy.

Reviewer #1: **Yes: **NAHYA SALIM MASOUD

Reviewer #2: No

---

## [Decision Letter · Decision Letter 2]

15 Nov 2023

PGPH-D-23-00351R2

Effectiveness of a multi-country implementation-focused network on quality of care: delivery of interventions and processes for improved maternal, newborn and child health outcomes

Dear Dr. Colbourn,

Thank you for submitting your manuscript to PLOS Global Public Health. After careful consideration, we feel that it has merit but does not fully meet PLOS Global Public Health’s publication criteria as it currently stands. Therefore, we invite you to submit a revised version of the manuscript that addresses the points raised during the review process.

Thank you for the well-written paper and for making revisions according to the reviewers' comments. There are still a few suggestions that I think the paper could benefit from. Please consider the following points and make appropriate changes before I accept the paper for publication.

The findings (in the Results and especially in the Abstract) seem to be quite similar in the four QCN focus countries. Were there any differences between the four countries? Was there cross-country learning (exchange of knowledge and experience)? This is important because the four countries have different health system governance and accountability, which is also reflected in the organisation of maternal and newborn care. For example, how did the governance and accountability system work in Ethiopia, given the highly decentralised health system and the health extension programme? How did it differ from other countries? how did this influence efforts to improve the quality of care? How did other countries learn from it?

More points along these lines would make the abstract more attractive. I have seen that most of the answers to these questions are there, but a bit scattered in the paper. It would be great to have a focused reflection to maximise the readership and impact of the paper.

We look forward to receiving your revised manuscript.

Kind regards,

Anteneh Asefa Mekonnen, Ph.D., MPH

Academic Editor

Journal Requirements:

Additional Editor Comments (if provided):

Reviewers' comments:

Reviewer's Responses to Questions

**Comments to the Author**

1. If the authors have adequately addressed your comments raised in a previous round of review and you feel that this manuscript is now acceptable for publication, you may indicate that here to bypass the “Comments to the Author” section, enter your conflict of interest statement in the “Confidential to Editor” section, and submit your "Accept" recommendation.

Reviewer #1: All comments have been addressed

2. Does this manuscript meet PLOS Global Public Health’s publication criteria? Is the manuscript technically sound, and do the data support the conclusions? The manuscript must describe methodologically and ethically rigorous research with conclusions that are appropriately drawn based on the data presented.

Reviewer #1: Yes

3. Has the statistical analysis been performed appropriately and rigorously?

Reviewer #1: Yes

4. Have the authors made all data underlying the findings in their manuscript fully available (please refer to the Data Availability Statement at the start of the manuscript PDF file)?

Reviewer #1: Yes

5. Is the manuscript presented in an intelligible fashion and written in standard English?

Reviewer #1: Yes

6. Review Comments to the Author

Reviewer #1: all comments responded, kudos to the authors

7. PLOS authors have the option to publish the peer review history of their article (what does this mean?). If published, this will include your full peer review and any attached files.

**Do you want your identity to be public for this peer review?** For information about this choice, including consent withdrawal, please see our Privacy Policy.

Reviewer #1: **Yes: **NAHYA SALIM MASOUD

---

## [Decision Letter · Decision Letter 3]

22 Jan 2024

Effectiveness of a multi-country implementation-focused network on quality of care: delivery of interventions and processes for improved maternal, newborn and child health outcomes

PGPH-D-23-00351R3

Dear Prof Colbourn,

We are pleased to inform you that your manuscript 'Effectiveness of a multi-country implementation-focused network on quality of care: delivery of interventions and processes for improved maternal, newborn and child health outcomes' has been provisionally accepted for publication in PLOS Global Public Health.

Best regards,

Julia Robinson

Executive Editor

Reviewer Comments (if any, and for reference):

Reviewer's Responses to Questions

**Comments to the Author**

1. If the authors have adequately addressed your comments raised in a previous round of review and you feel that this manuscript is now acceptable for publication, you may indicate that here to bypass the “Comments to the Author” section, enter your conflict of interest statement in the “Confidential to Editor” section, and submit your "Accept" recommendation.

Reviewer #1: All comments have been addressed

2. Does this manuscript meet PLOS Global Public Health’s publication criteria? Is the manuscript technically sound, and do the data support the conclusions? The manuscript must describe methodologically and ethically rigorous research with conclusions that are appropriately drawn based on the data presented.

Reviewer #1: Yes

3. Has the statistical analysis been performed appropriately and rigorously?

Reviewer #1: Yes

4. Have the authors made all data underlying the findings in their manuscript fully available (please refer to the Data Availability Statement at the start of the manuscript PDF file)?

Reviewer #1: Yes

5. Is the manuscript presented in an intelligible fashion and written in standard English?

Reviewer #1: Yes

6. Review Comments to the Author

Reviewer #1: The manuscript is recommended to proceed with final checks

7. PLOS authors have the option to publish the peer review history of their article (what does this mean?). If published, this will include your full peer review and any attached files.

**Do you want your identity to be public for this peer review?** For information about this choice, including consent withdrawal, please see our Privacy Policy.

Reviewer #1: **Yes: **NAHYA SALIM MASOUD
